# AAV-Mediated Targeting of the Activin A-ACVR1^R206H^ Signaling in Fibrodysplasia Ossificans Progressiva

**DOI:** 10.3390/biom13091364

**Published:** 2023-09-08

**Authors:** Yeon-Suk Yang, Chujiao Lin, Hong Ma, Jun Xie, Frederick S. Kaplan, Guangping Gao, Jae-Hyuck Shim

**Affiliations:** 1Department of Medicine, Division of Rheumatology, University of Massachusetts Chan Medical School, Worcester, MA 01655, USA; yen.yang@umassmed.edu (Y.-S.Y.); chujiao.lin@umassmed.edu (C.L.); 2Horae Gene Therapy Center, University of Massachusetts Chan Medical School, Worcester, MA 01655, USA; hong.ma2@umassmed.edu (H.M.); jun.xie@umassmed.edu (J.X.); 3Department of Microbiology and Physiological Systems, University of Massachusetts Chan Medical School, Worcester, MA 01655, USA; 4Viral Vector Core, University of Massachusetts Chan Medical School, Worcester, MA 01655, USA; 5Department of Orthopaedic Surgery, The Perelman School of Medicine at the University of Pennsylvania, Philadelphia, PA 19104, USA; frederick.kaplan@pennmedicine.upenn.edu; 6Department of Medicine, The Perelman School of Medicine at the University of Pennsylvania, Philadelphia, PA 19104, USA; 7The Center for Research in FOP and Related Disorders, The Perelman School of Medicine at the University of Pennsylvania, Philadelphia, PA 19104, USA; 8Li Weibo Institute for Rare Diseases Research, University of Massachusetts Chan Medical School, Worcester, MA 01655, USA

**Keywords:** fibrodysplasia ossificans progressiva, heterotopic ossificans, Activin A, ACVR1, AAV, gene therapy

## Abstract

Fibrodysplasia ossificans progressiva (FOP) is an ultra-rare genetic disorder characterized by progressive disabling heterotopic ossification (HO) at extra-skeletal sites. Here, we developed adeno-associated virus (AAV)-based gene therapy that suppresses trauma-induced HO in FOP mice harboring a heterozygous allele of human *ACVR1^R206H^* (*Acvr1^R206H/+^*) while limiting the expression in non-skeletal organs such as the brain, heart, lung, liver, and kidney. AAV gene therapy carrying the combination of codon-optimized human ACVR1 (ACVR1^opt^) and artificial miRNAs targeting Activin A and its receptor ACVR1^R206H^ ablated the aberrant activation of BMP-Smad1/5 signaling and the osteogenic differentiation of *Acvr1^R206H/+^* skeletal progenitors. The local delivery of AAV gene therapy to HO-causing cells in the skeletal muscle resulted in a significant decrease in endochondral bone formation in *Acvr1^R206H/+^* mice. These mice showed little to no expression in a major AAV-targeted organ, the liver, due to liver-abundant miR-122-mediated repression. Thus, AAV gene therapy is a promising therapeutic strategy to explore in suppressing HO in FOP.

## 1. Introduction

Fibrodysplasia ossificans progressiva (FOP, #OMIM 135100) is an ultra-rare genetic disorder that causes the progressive disabling heterotopic ossification (HO) of connective tissues, such as muscle, ligaments, tendons, fascia, and aponeuroses. HO is formed through endochondral ossification in childhood through adulthood, leading to immobility and severe pain [1]. Flare-ups occur spontaneously in connective tissues or are triggered by injury, immunization via intramuscular injection, inflammation, or unknown reasons, followed by heterotopic bone formation. As a primary symptomatic management, high-dose corticosteroids for episodic flare-ups is used to decrease symptoms, including pain and edema [2]. The early onset of extra-skeletal formation and the difficulty in selectively inhibiting the aberrant activation of *ACVR1^R206H^* signaling make the treatment and prevention of FOP challenging. Currently, a retinoic acid receptor γ agonist (palovarotene capsules, Sohonos) was approved for treating FOP in Canada and USA but not in Europe, while an anti-Activin A antibody (REGN 2477), an immunosuppressant (rapamycin), and ACVR1 kinase inhibitors (IPN60130, Saractinib, INCB000928) are in clinical trials [3]. 

Recombinant adeno-associated viruses (rAAVs) are highly effective in transducing the liver, skeletal muscle, and the skeleton in vivo [4], as well as the long-term expression of therapeutic gene [5] and the robust safety profiles in both pre-clinical and clinical studies. Since gene therapy using rAAVs holds promise for treating many mono-genetic disorders, AAV gene therapy might be an attractive therapeutic approach for FOP in that approximately 97% of FOP patients harbor a recurrent, missense *ACVR1^R206H^* mutation (c.617G>A;p.R206H) [6]. Our previous study demonstrated that the AAV9 serotype is effective for the transduction of fibro-adipogenic progenitors (FAPs), major cells-of-origin of HO [7,8]. Treatment with the AAV9 vector carrying *ACVR1^R206H^*-specific silencer and codon-optimized human ACVR1 reduced the aberrant activation of bone morphogenetic protein (BMP)-Smad1/5 signaling and the chondrogenic/osteogenic differentiation of *Acvr1^R206H^* skeletal progenitors. Accordingly, the systemic delivery of AAV gene therapy prevented spontaneous HO in FOP mice while trauma-induced HO was also decreased when administered transdermally to injured muscle [9]. However, since a high dose administration of rAAVs and/or AAV expression in non-HO tissues could potentially induce untoward immune responses and side effects in FOP patients, further vector improvement to reduce an injection dose by enhancing therapeutic efficacy and repress AAV expression in non-HO tissues is needed. 

The *ACVR1^R206H^* mutation has been reported to confer neoactivity to Activin A, which confers the aberrant activation of BMP-pSmad1/5 signaling [10,11]. This suggests that Activin A functions as a gain-of-function ligand of the ACVR1^R206H^ receptor in FOP [12,13]. In this study, we sought to improve therapeutic efficacy of AAV vector by blocking the expression of the mutant ACVR1^R206H^ receptor and replacing it with WT ACVR1 receptors while also silencing Activin A expression simultaneously [9,14]. The inhibition of both Activin A and its receptor ACVR1^R206H^ ablated aberrant BMP-Smad1/5 activation and the osteogenic differentiation of *Acvr1^R206H^* skeletal progenitors in response to Activin A. Additionally, the local delivery of the AAV gene therapy to injured muscle was highly effective in both the prevention and treatment of trauma-induced HO in mice harboring a heterozygous allele of human *ACVR1^R206H^* (*Acvr1^R206H/+^*). Finally, the safety of AAV gene therapy was further improved by selectively repressing the expression in the liver via the liver-abundant miR-122-mediated degradation. Thus, this proof-of-concept study using AAV gene therapy provides insights into clinical translation for FOP patients.

## 2. Materials and Methods

### 2.1. Mice

Mice harboring a knock-in allele of human *Acvr1^R206H^* (*Acvr1^(R206H)Fl^*) [15] were obtained from the International FOP Association (IFOPA) via Dr. Daniel Perrien (Emory University) and maintained on a C57BL/6J background. The target construct was designed to delete mouse exons 5–10, neomycin-resistant genes (neo cassette), one F3 site, and one loxP site in the presence of Cre recombinase, thereby expressing human cDNA exons 6–11 containing the R206H mutation and eGFP. The target construct was inserted into the locus of mouse *Acvr1* gene (Appendix A). *Acvr1^(R206H)Fl^* mice were crossed with *Sox2-Cre* mice (Jackson laboratory, C57BL/6J), where the expression of Cre recombinase in epiblasts mediates the expression of *Acvr1^R206H^* in all tissues (*Acvr1^(R206H)Fl/+^;Sox2-cre*, hereafter referred to as *Acvr1^R206H/+^*), and *Prrx1-Cre* mice (Jackson laboratory, C57BL/6J) expressing Cre recombinase in Prrx1^+^ skeletal progenitors in the limb mesenchyme mediate *Acvr1^R206H^*-driven HO (*Acvr1^(R206H)Fl^;Prrx1-cre,* hereafter referred to as *Acvr1^R206H^;Prx1*). *Acvr1^R206H/+^* mice were crossed with *Pdgfrα-GFP* reporter mice (Jackson Laboratory, C57BL/6J) to label Pdgfrα^+^ FAPs. A polymerase chain reaction (PCR) assay of tail genomic DNA was used to determine mouse genotypes. Littermate controls were used for all experiments. All animals were used in accordance with the NIH Guide for the Care and Use of Laboratory Animals and were handled according to protocols approved by the University of Massachusetts Chan Medical School Institutional Animal Care and Use Committee (IACUC). 

### 2.2. Cell Culture and Reagents

Human bone marrow-derived mesenchymal stromal cells (BMSCs, Cat. #7500), purchased from ScienCell Research Laboratories (Carlsbad, CA, USA), were cultured according to the manufacturers’ manuals. GFP^+^ScaI^+^CD31^−^CD45^−^ FAPs were FACS sorted from the digested skeletal muscle of 8-week-old *Acvr1^R206H/+^;PDGFR*α-GFP mice using cell surface markers (PDGFRα-GFP^+^Sca1^+^CD31^−^CD45^−^) [7] and their frequency in the injured muscle were assessed using flow cytometry. Alternatively, they were FACS-sorted from the injured muscle and subjected to RT-PCR analysis. Mouse BMSCs were obtained from the long bones of 4-week-old *Acvr1*^(*R206H*)*Fl*^*;PRRX1-Cre* mice and cultured in α-MEM medium (Gibco) containing 10% FBS (Corning), 2 mM L-glutamine (Corning), 1% penicillin/streptomycin (Corning), and 1% nonessential amino acids (Corning). Alternatively, they were differentiated into osteoblasts under osteogenic medium containing ascorbic acid (200 uM, MilliporeSigma, Burlington, VT, USA, Cat #: A8960) and β-glycerophosphate (10 mM, Sigma, #G9422). Recombinant lipopolysaccharide (Cat #: LPS-B5), TNF (Cat #: 210-TA), IL-1β/IL-1F2 (Cat #: 201-LB), and Activin A (Cat #: 338-AC) proteins were purchased from InvivoGen (San Diego, CA, USA) and R&D systems (Minneapolis, MN, USA), respectively. Plasmids expressing HA-tagged human or mouse Inhba were purchased from Sino Biological (Beijing, China).

### 2.3. rAAV Vector Design and Production

To generate the AAV vector genome for the FOP gene therapy, first, a codon-optimized version of the human ACVR1 complementary DNA (*ACVR1^opt^*) was cloned into the pAAVsc-CB6 plasmid by replacing the *mCherry* reporter with *ACVR1^opt^*. Second, the chicken β actin (CBA) intron in the plasmid was replaced with a 384 bp MBL intron (developed by MassBiologics, Boston, MA, USA) [9]. A custom Excel macro, which considers miR-33 scaffold design rules to generate optimized artificial miRNAs (amiR) cassettes [16], was used to design human *ACVR1^R206H^* amiR or mouse/human *Inhba* aimRs. Plasmids were constructed via Gibson assembly and standard molecular biology methods. gBlocks for *amiR-33-ctrl*, *amiR-33-human ACVR1^R206H^*, and *amiR-33-Inhba* were synthesized and inserted into the intronic region of the pAAVsc-*CB6-mCherry* plasmid at the restriction enzyme sites (PstI and BglII, Appendix A) [9]. After being verified through sequencing, the AAV vector genomes were packaged into an AAV9 capsid. rAAV production was executed via transient transfection in HEK293 cells, purified using CsCl sedimentation, and quantified via droplet digital PCR (ddPCR) on a QX200 ddPCR system (Bio-Rad, Hercules, CA, USA) using the *Egfp* or *mCherry* prime/probe set [16]. The sequences of gBlocks and ddPCR primers are listed in Appendix A. 

### 2.4. Quantitative RT-PCR, Immunoblotting, and ELISA Analyses

The total RNA was extracted from cells and/or muscle tissues using QIAzol (QIAGEN, Hilden, Germany) and then synthesized to cDNA using the High-Capacity cDNA Reverse Transcription Kit (Applied Biosystems, Waltham, MA, USA, Cat #: 4368814). Quantitative real time-PCR (qRT-PCR) was performed using iTaq™ Universal SYBR^®^ Green Supermix (Bio-Rad, Cat #: 1725122) on the CFX connect RT-PCR detection system (Bio-Rad). To measure the mRNA levels of the indicated genes at the injured sites or HO lesions, tibial muscles were snap-frozen in liquid nitrogen for 30 s and then homogenized in QIAzol for 1 min using the Beadbug microtube homogenizer. 

Cells were lysed in TNT lysis buffer (150 mM NaCl, 1% Triton X-100, 1 mM EDTA, 1 mM EGTA, 50 mM NaF, 1 mM Na_3_VO_4_, 1 mM PMSF, and protease inhibitor cocktail (MilliporeSigma)), and DC protein assay (Bio-Rad) was used to measure protein amounts from cell lysates. Equivalent amounts of proteins were loaded to SDS-PAGE gels, transferred to Immobilon-P membranes (MilliporeSigma), immunoblotted with anti-phospho-Smad1/5 antibodies (1:1000, Cell Signaling Technology, Danvers, MA, USA, Cat #: 9516) and anti-GAPDH antibodies (1:1000, MilliporeSigma, #CB1001), and developed using Pierce ECL Western Blotting Substrate (ThermoFisher Scientific, Waltham, MA, USA). Immunoblotting with anti-HSP90 antibody was used as a loading control. 

AAV-treated muscles were lysed using NP-40 lysis buffer (ThermoFisher Scientfic, #J60766) and protein levels of Activin A in the lysates were measured using the Activin A Quantikine ELISA kit (R&D systems, #DAC00B). Alternatively, the supernatant was harvested from AAV-treated *Acvr1^R206H^* BMSCs and then subjected to the Activin A ELISA kit. 

### 2.5. Trauma-Induced HO

Prevention model: A total of 5 × 10^12^ vg/kg of rAAV9.*gfp.miR-122.TS* or rAAV9.*amiR-Inhba/Acvr1^R206H^.ACVR1^opt^.miR-122.TS* was injected transdermally (t.d.) into the tibial muscle of 8-week-old *Acvr1^R206H/+^* mice. Three days later, a pinch injury was employed in the gastrocnemius muscle. mRNA levels of *Inhba*, *ACVR1^R206H^* and *ACVR1^opt^* were assessed using RT-PCR four weeks post-injury. Heterotopic bone mass in the muscle was assessed via radiography, and a microCT was performed four weeks post-injury. Alternatively, pinch injury was utilized on the gastrocnemius muscle of AAV-treated *Acvr1^R206H/+^;Pdgfrα-GFP* mice and four weeks later, AAV-transduced FAPs were FACS sorted from HO sites using GFP^+^Sca1^+^CD31^−^CD45^−^ markers. GFP-expressing FAPs in the cryosectioned muscle were visualized using fluorescence microscopy. Treatment model: 8-week-old *Acvr1^R206H/+^* tibial muscles were injected t.d. with rAAV9.*gfp.miR-122-TS* or rAAV9.*amiR-Inhba/Acvr1^R206H^.ACVR1^opt^.miR-122-TS* one, three, or six days after the pinch injury. Four weeks later, mice were euthanized for HO assessment.

### 2.6. Osteoblast Differentiation

To stain alkaline phosphatase (ALP) on the surface of osteoblasts, cells were fixed with 10% neutral buffered formalin and then stained with the ALP staining solution containing Fast Blue (MilliporeSigma, Cat #: FBS25) and Naphthol AS-MX (MilliporeSigma, Cat #: 855). Alternatively, to measure cell proliferation rate, osteoblasts were incubated with 10-fold diluted Alamar Blue solution (Invitrogen, Waltham, MA, USA, Cat #: DAL1100) for 1 h and then measured using a fluorescence plate reader (Molecular Devices, San Jose, CA, USA). After washing with cold PBS, cells were incubated with an ALP solution containing 6.5 mM Na_2_CO_3_, 18.5 mM NaHCO_3_, 2 mM MgCl_2_, and phosphatase substrate (MilliporeSigma, Cat #: S0942), and ALP activity was measured via a spectrometer (BioRad).

For extracellular matrix mineral staining, osteoblasts were fixed in 70% EtOH for 15 min at room temperature, washed with distilled water, and stained with 2% alizarin red solution (MilliporeSigma, Cat #: A5533) for 15 min. After washing with distilled water, alizarin red-stained calcium deposits were quantified using the acetic acid extraction method [17]. 

### 2.7. MicroCT and Radiography

The MicroCT scanner (uCT35; SCANCO Medical AG, Wangen-Brüttisellen, Switzerland) was used for the qualitative and quantitative assessment of heterotopic bone in muscle and performed by an investigator blinded to the genotypes of the animals under analysis. MicroCT scanning was conducted at 55 kVp and 114 mA energy intensity with an integration time of 300 ms. The nominal voxel size used for whole tibia was 12 μm. All images were reconstructed using image matrices of 1024 × 1024 pixels. For heterotopic bone evaluation, the whole tibia muscle area was contoured. Three-dimensional reconstruction images were generated from contoured two-dimensional images using the distance transformation of the binarized images. The Inveon multimodality 3D visualization program was also used to generate 3D reconstruction images of multiple static or dynamic volumes of microCT modalities (Siemens Medical Solutions, Malvern, PA, USA). All images presented are representative of the respective genotypes (n > 5).

The Trident Specimen Radiography System (Hologic, Marlborough, MA, USA) was used to generate radiographic images of the whole body of AAV-treated mice. An X-ray beam intensity of 1 mA 28~30 KV with automatic exposure control (AEC) was used for fast image acquisition.

### 2.8. Histology

The whole hindlimb with intact skeletal muscle was dissected from mice, fixed in 10% neutral buffered formalin for two days at room temperature, and then decalcified with 14% EDTA tetrasodium salt (pH 7.6) for 3–4 weeks in a cold room. Samples were stored in 70% ethanol then processed on a vacuum infiltration tissue processor (Leica, Teaneck, NJ, USA). A microtome (HistoCore Multicut, Leica) was used to perform serial sections at a thickness of 6 μm along the coronal plate from anterior to posterior and then slides were subjected to hematoxylin and eosin (H&E) or alcian blue/hematoxylin/orange G staining.

For tissue cryo-sectioning, dissected tissues were fixed in 4% paraformaldehyde for 2~3 days and then decalcified with 14% EDTA tetrasodium salt (pH 7.6) for 10 days in a cold room. A 20% sucrose solution was used for infiltration prior to OCT embedding. A cryostat (LM3050s, Leica) was used to perform serial sections at a thickness of 12 μm, and then slides were stained with DAPI or H&E.

### 2.9. Statistical Analysis

Animals were randomized to AAV treatment vs. control groups and all animals or samples were included for statistical analysis. Sample sizes were determined based on a 30% difference in the parameters measured, which is considered biologically significant with 10–20% of the expected mean sigma value. The mean ± SD was used for statistical analysis and Alpha and Beta were set to the standard values of 0.05 and 0.8, respectively. For relevant data analysis, the Shapiro–Wilk normality test was performed to check normal distributions of the groups. If the normality tests were passed, two-tailed, Student’s unpaired *t*-test was used, and if the normality tests were failed, and Mann–Whitney tests were used for the comparisons between two groups. One-way ANOVA was used to compare three or four groups and if the normality were tests passed, Tukey’s multiple comparison test for all pairs of groups was performed. If the normality tests failed, the Kruskal–Wallis test and Dunn’s multiple comparison test were performed. We used the GraphPad PRISM software version 10 for statistical analysis. *p* < 0.05 was considered statistically significant: *, *p* < 0.05; **, *p* < 0.01; ***, *p* < 0.001; and ****, *p* < 0.0001.

## 3. Results

### 3.1. Upregulation of Activin A in HO-Tissues of FOP Mice

Activin A is a member of the TGF-β superfamily and plays pivotal roles in the regulation of tissue homeostasis, organ development, inflammation, cell proliferation, and apoptosis [18]. Activin A expression is rapidly upregulated by inflammatory macrophages and other activated immune cells, which leads to the production of pro-inflammatory cytokines, such as TNF, IL-1β, and IL-6, and the recruitment of mast cells, thereby initiating HO pathogenesis [19]. Using single cell RNA-seq analysis and developmental trajectories, a recent study demonstrated that in addition to inflammatory immune cells, Sox9^+^ mesenchymal and chondrogenic progenitors also expressed Activin A in BMP2-induced HO tissues [20]. This is consistent with our findings that Activin A expression was markedly increased in the established HO tissues within the injured muscles of *Acvr1^R206H/+^* mice compared to non-HO tissues in contralateral legs (Figure 1A). Likewise, elevated levels of Activin A were also detected in bone marrow-derived stromal cells (BMSCs) isolated from FOP mice (*Acvr1^R206H^;Prx1*) relative to WT mice (*Acvr1^+/+^;Prx1,*
Figure 1B), indicating that heterotopic bone-forming, chondrogenic/osteogenic cells are also Activin A-producing cell populations in FOP. Notably, human BMSCs were well responsive to osteogenic cues (Figure 1C) or pro-inflammatory stimuli, including lipopolysaccharide (LPS), tumor necrosis factor (TNF), or interleukin 1β (IL-1β, Figure 1D), resulting in the upregulation of Activin A. Similarly, levels of Activin A in mouse *Acvr1^R206H^;Prx1* BMSCs were highly upregulated via treatment with proinflammatory stimuli relative to PBS (Figure 1E). Thus, these results suggest that chondrogenic/osteogenic cells may facilitate HO development by producing Activin A under flare-up conditions.

### 3.2. Generation of the AAV Gene Therapy Targeting Activin A

To examine whether the upregulation of Activin A in *Acvr1^R206H^;Prx1* BMSCs through pro-inflammatory stimuli can be reversed by replacing the mutant ACVR1^R206H^ receptor with wildtype (WT) ACVR1 receptor, *Acvr1^R206H^;Prx1* BMSCs were transduced with the AAV vector carrying control (amiR-*ctrl*) or the combination of artificial miRNA targeting *ACVR1^R206H^* mRNA (amiR-*ACVR1^R206H^*) and codon-optimized human ACVR1 (*ACVR1^opt^*) [9] and then stimulated with LPS, TNF, or IL-1β. An ELISA analysis showed that Activin A expression levels were comparable between amiR-ctrl- and amiR-ACVR1^R206H^.ACVR1^opt^-treated cells (Figure 1E), suggesting that the expression of ACVR1^R206H^ receptor in BMSCs is dispensable to the upregulation of Activin A. Thus, we generated the AAV vector, silencing the expression of both Activin A and the ACVR1^R206H^ receptor to inhibit the Activin A-ACVR1^R206H^ signaling pathway more effectively. Given that Activin A is an homodimers of Inhibins βA (Inhba) [18], six amiRs targeting the shared coding sequences of mouse and human *Inhba* (amiR-Inhba #1–#6) were designed to silence the expression of both mouse and human Activin A. Either amiR-ctrl or amiR-Inhba #1–#6 were co-transfected with an HA-tagged mouse or human Inhba plasmid (Inhba-HA) in HEK293 cells and then the protein levels of Inhba-HA in cell lysates were assessed through immunoblotting with the anti-HA antibody (Figure 2A). These results demonstrated that amiR-Inhba #4 and #6 with full homology to mouse and human *Inhba* effectively downregulated the expression of both mouse and human Inhba. In particular, the AAV9 carrying amiR-Inhba #4 was highly effective for the silencing of both mouse and human *Inhba* expression in *Acvr1^R206H^;Prx1* BMSCs and human BMSCs, respectively (Figure 2B; hereafter referred to amiR-Inhba). These cells also showed a significant decrease in the upregulation of Activin A via pro-inflammatory stimuli (Figure 2C). Therefore, amiR-Inhba #4 is a potent gene silencer that can suppress Activin A expression.

### 3.3. Generation of the AAV Gene Therapy with Liver-Specific Repression

Our previous studies demonstrated that a single dose (5 × 10^13^ vg/kg) of systemically delivered AAV9 transduced liver, heart, skeletal muscle, and bone in adult mice [4] while skeletal muscle was transduced when transdermally (t.d.) administered at 5 × 10^12^ vg/kg [9]. Notably, a tissue distribution analysis in mice using the IVIS optical imaging system also revealed a robust expression of t.d. injected AAV9 in the liver due to the high trans-vascular efficiency of the AAV9 capsid [9,21]. We, therefore, repressed AAV9′s expression in the liver via liver-abundant miR-122-mediated degradation to improve the safety of the AAV gene therapy. Given that the silencing of transgene expression in liver exploited the natural expression of the abundant (≥60,000 copies/cell) miRNA, miR-122, in hepatocytes of virtually all animals [22,23], endogenous complementary sites for miR-122 (miR-122-TS) were inserted into the 3′ untranslated region (UTR) of *gfp* or *ACVR1^opt^* transgene in the AAV vector genome and packaged into AAV9 capsid (AAV9.gfp.MIR, AAV9.Acvr1/Inhba.MIR). IVIS optical imaging system revealed that the t.d. injection of AAV9.gfp.MIR successfully expressed GFP in the skeletal muscle while little to no expression was detected in the brain, heart, lung, liver, kidney, or spleen (Figure 3A). This is consistent with fluorescence microscopy showing GFP expression in the skeletal muscle but not in brain, heart, or liver (Figure 3B). Thus, the local delivery of our AAV gene therapy with liver-abundant miR-122-mediated repression enables the detargeting of non-skeletal organs except for skeletal muscle.

### 3.4. AAV Gene Therapy Inhibits Aberrant ACVR1^R206H^ Signaling and Osteogenic Differentiation of Acvr1^R206H^ Skeletal Progenitors

The liver-detargeting AAV vector genome that silences the expression of Activin A and ACVR1^R206H^ receptor and expresses ACVR1^opt^ receptor simultaneously was generated and packaged into an AAV9 capsid (hereafter referred to as AAV9.Acvr1/Inhba.MIR, Figure 4A). To examine the in vitro therapeutic efficacy of AAV9 carrying amiR-ACVR1^R206H^.amiR-Inhba.ACVR1^opt^.MIR (Acvr1/Inhba), *Acvr1^R206H^;Prx1* BMSCs were transduced by the AAV vectors, and six days later, the knockdown efficiency of *Inhba* or *ACVR1^R206H^* and the expression of *ACVR1^opt^* were confirmed via RT-PCR analysis (Figure 4B). The osteogenic differentiation of AAV-transduced BMSCs were examined, demonstrating a significant decrease in alkaline phosphatase (ALP) activity (Figure 4C) and the expression of osteogenic genes, including *Tnalp, Osterix* (*Osx*), *Bone sialoprotein* (*Ibsp*), and *Osteocalcin* (*Bglap*, Figure 4D). Additionally, Activin A-induced ALP activity and the extracellular mineralization of these cells were substantially reduced through treatment with Acvr1/Inhba relative to the control (Figure 4E). This is accompanied by a significant decrease in the induction of BMP-responsive genes, *Id1* and *Msx2* (Figure 4F), and the phosphorylation of BMP signaling mediators, Smad1 and 5, in response to Activin A (Figure 4G). Thus, the AAV gene therapy of silencing *Inhba* and *ACVR1^R206H^* expression and expressing the ACVR1^opt^ receptor is a potent inhibitor of Activin A-induced aberrant BMP signaling and osteogenic differentiation in FOP skeletal progenitors.

### 3.5. AAV Gene Therapy Prevents Trauma-Induced HO in FOP Mice

Since the constitutive expression of human *ACVR1^R206H^* allele causes perinatal lethality in mice [24], mice harboring a conditional knock-in allele of human *ACVR1^R206H^* (*Acvr1^(R206H)Fl^*) were crossed with Sox2-Cre mice (*Acvr1^(R206H)Fl/+^;Sox2-cre,* hereafter referred to *Acvr1^R206H/+^*) where the expression of Cre recombinase in epiblasts mediates the expression of *Acvr1^R206H^* in all tissues (Appendix A). Heterozygous mice (*Acvr1^R206H/+^*) were further crossed with *Pdgfrα-GFP* reporter mice to label Pdgfrα-expressing FAPs, the major cells-of-origin of HO (*Acvr1^R206H/+^;Pdgfrα-GFP*) [7,8]. To examine the ability of AAV gene therapy to prevent trauma-induced HO in FOP, 8-week-old *Acvr1^R206H/+^;Pdgfrα-GFP* mice were treated with a single dose of a t.d. injection of AAV9.ctrl.MIR or AAV9.Acvr1/Inhba.MIR, and three days later, a pinch injury was introduced into the gastrocnemius muscle (Figure 5A). Four weeks after the injury, Pdgfrα-expressing FAPs were isolated from the treated muscles via FACS sorting using GFP^+^ScaI^+^CD31^−^CD45^−^ markers, and the knockdown efficiency of *ACVR1^R206H^* or *Inhba or ACVR1^opt^* expression in these cells was validated using RT-PCR analysis (Figure 5B). Radiography (Figure 5C) and microCT (Figure 5D) analyses demonstrated that the heterotopic bone mass in the injured muscle was substantially decreased when treated with Acvr1/Inhba.MIR relative to ctrl.MIR. Thus, the local delivery of AAV gene therapy targeting both Activin A and the ACVR1^R206H^ receptor is highly effective for the prevention of trauma-induced HO in the skeletal muscle of *Acvr1^R206H/+^* mice.

To visualize how Acvr1/Inhba.MIR expression prevents the pathogenesis of HO in FOP mice, a pinch injury was introduced in the gastrocnemius muscle of AAV-treated *Acvr1^R206H/+^;Pdgfrα-GFP* mice, and four weeks later, Pdgfrα-expressing FAPs in the injured muscle were monitored via fluorescence microscopy using GFP expression (Figure 5E, left). As expected, GFP-expressing Pdgfrα^+^ FAPs within the forming HO lesions in ctrl-treated muscle were highly proliferative, and a subset of the cells were differentiated into heterotopic bone-forming osteoblasts. This was consistent with the histologic analysis showing heterotopic bone and chondrogenic anlagen in the skeletal muscle of these mice (Figure 5E, right). By contrast, there was little to no evidence of HO lesions in the skeletal muscle expressing Acvr1/Inhba, where GFP-expressing Pdgfrα^+^ FAPs were primarily present in muscle interstitium (Figure 5E), similar to Pdgfrα^+^ FAPs present in wildtype muscle [9]. Likewise, a flow cytometry analysis demonstrated a significant decrease in the numbers of GFP^+^ScaI^+^CD31^−^CD45^−^ FAPs in the skeletal muscle treated with Acvr1/Inhba relative to ctrl (Figure 5F). Notably, these cells showed a significant reduction in *Id1* (BMP-responsive gene), *Sox9* (chondrogenic gene), and *Col1α1* (osteogenic gene) expression (Figure 5G), suggesting that the local delivery of AAV9.Acvr1/Inhba.MIR effectively suppresses ACVR1^R206H^-induced aberrant BMP signaling and resultant chondrogenesis and osteogenesis of Pdgfrα^+^ FAPs in *Acvr1^R206H/+^* mice. Thus, the inhibition of the Activin A and ACVR1^R206H^ receptor in the skeletal muscle of FOP mice using AAV9.Acvr1/Inhba.MIR is a promising approach to inhibit the initiation process of traumatic HO in FOP.

### 3.6. Treatment of Traumatic HO in FOP Mice through AAV Gene Therapy

Previous studies demonstrated that upon muscle injury, FOP mice undergo HO lesion progression from immune cell infiltration (Day 1–3), muscle degeneration and fibroproliferative response (Day 3–7), chondrogenesis (Day 7–14), through osteogenesis with heterotopic bone marrow establishment (Day 14–28) [25]. To define the stage at which AAV gene therapy can suppress HO progression, a pinch injury was introduced into the gastrocnemius muscle of 8-week-old *Acvr1^R206H/+^* mice, and one, three, or six days later, AAV9 carrying ctrl or Acvr1/Inhba was t.d. injected (Figure 6A). Four weeks later, the knockdown efficiency of *ACVR1^R206H^* or *Inhba* and the expression of *ACVR1^opt^* in treated muscles was validated using RT-PCR analysis (Figure 6B). Radiography, microCT, and the histopathological evaluation of AAV-treated muscle was performed, demonstrating that heterotopic bone mass was markedly decreased by the local delivery of AAV9.Acvr1/Inhba.MIR at different time points (Figure 6C–E). As expected, ctrl-treated muscle developed heterotopic bone with a bone marrow compartment via endochondral pathway, including the formation of fibrotic tissue and chondrogenic analgen. However, treatment with AAV9.Acvr1/Inhba.MIR day 1 or 3 post-injury resulted in a significant decrease in the volume of heterotopic bone, which was filled with adipose tissue (inside) and surrounded by regenerated muscle (outside). When treated with AAV9.Acvr1/Inhba.MIR on day 6 post-injury, small-sized heterotopic bones with bone marrow compartments (inside) were formed and surrounded by regenerated muscle (outside, Figure 6E). These results demonstrate that treatment with AAV gene therapy before the chondrogenic stage effectively suppresses HO progression. Taken together, an AAV-mediated combination gene therapy that executes the silencing of *Inhba* and *ACVR1^R206H^* and *ACVR1^opt^* expression is a promising approach to suppress disabling HO, providing proof-of-concept for clinical translation to FOP patients.

## 4. Discussion

Our data demonstrate the AAV gene therapy targeting Activin A and its receptor ACVR1^R206H^ as a potent suppressor of HO in FOP. Here, we developed a multi-functional AAV vector which executes both the silencing of Activin A and ACVR1^R206H^ and the expression of codon-optimized human ACVR1 (ACVR1^opt^), while detargeting the liver. As a key stimulator of HO, Activin A is upregulated during the chondrogenic/osteogenic differentiation of FOP skeletal progenitors in HO tissues, and our AAV gene therapy downregulated the production of Activin A in these cells. Thus, AAV treatment ablated the aberrant activation of BMP-Smad1/5 signaling and the osteogenic differentiation of *Acvr1^R206H/+^* skeletal progenitors. Additionally, the local delivery of AAV gene therapy to the skeletal muscle before or after pinch injury markedly decreased endochondral bone formation in *Acvr1^R206H/+^* mice, suggesting AAV gene therapy as a potent suppressor that can prevent or treat trauma-induced HO in FOP. Finally, the insertion of target sequences for liver abundant miR-122 into AAV vector genome enabled the repression of AAV9′s expression in the liver. Thus, our AAV gene therapy is an attractive therapeutic strategy to suppress HO in FOP patients who have a lifelong progression of HO. 

Currently FDA-approved AAV gene therapy for lipoprotein lipase deficiency (Glybera) [26], inherited retinal dystrophy (Luxturna) [27], spinal muscular atrophy (Zolgensma) [28], Duchenne muscular dystrophy (Elevidys) [29], and hemophilia B (Hemgenix) [30] utilizes a gene replacement strategy that introduces genes encoding missing proteins or encoding corrective proteins. On the other hand, since autosomal dominant, heterozygous mutation in *ACVR1^R206H^* (c.617G>A;p.R206H) causes FOP [6,31], our gene therapy approach was designed to replace the mutant ACVR1^R206H^ receptor with the healthy ACVR1 receptor via the combination approach: (1) the silencing of the expression of the ACVR1^R206H^ receptor at the mRNA level using *ACVR1^R206H^* allele-specific amiR and (2) the dilution of aberrant BMP-Smad1/5 signaling by the mutant ACVR1^R206H^ receptor with the codon-optimized WT ACVR1 receptor. Additionally, its therapeutic efficacy was further enhanced by the silencing expression of the key HO inducer Activin A at the mRNA level using amiR-Inhba. Our results provide a proof-of-concept demonstration that the AAV-mediated inhibition of both Activin A and its receptor ACVR1^R206H^ in injured muscle is highly effective in suppressing the initiation and progression of HO in FOP mice. 

Since the immunization of FOP patients via i.m. injection often induces muscle trauma and a flare-up, followed by the formation of HO lesions, we delivered AAV gene therapy to skeletal muscle via t.d. injection using a hollow microneedle to avoid muscle trauma from i.m. injection [32]. Consistent with our previous study demonstrating that the t.d. injection of AAV9 transduces Tie2^+^ or PDGFRα+ FAP-lineage cells in HO lesions [9] and that the local delivery of our AAV gene therapy to skeletal muscle via t.d. injection was effective for the transduction of FAP-lineage cells and suppressed heterotopic bone formation in the injured muscle of *Acvr1^R206H/+^* mice. Thus, the t.d. injection of AAV gene therapy might be a promising approach for treating trauma- and/or flare-up-induced HO in FOP patients. However, since the expression of t.d. injected AAV9 was also detected in the liver due to the high trans-vascular efficiency of AAV9 capsid [9], we repressed its expression in the liver via liver-abundant miR-122-mediated degradation since liver function in FOP patients is relatively normal. However, further studies will be necessary to examine the therapeutic outcomes of locally or systemically delivered AAV gene therapy at different doses in suppressing both traumatic and spontaneous HO in *Acvr1^R206H/+^* mice. Furthermore, the long-term durability and safety of therapeutic gene expression are crucial to consider in terms of the potential use of AAV gene therapy in FOP patients who are sensitive to the viral stimulation of pre-osseous lesions and need lifelong HO suppression. Finally, since inflammation can pose a high risk for HO development in FOP patients, any consideration of AAV gene therapy needs to be carefully approached. Although AAVs have low post-infection immunogenicity [21,33] and FOP mice showed little to no immune responses to systemically delivered AAV vectors [9], a high dosage of AAV could potentially induce immune responses in FOP patients, which may compromise the subsequent safety of this method as well as any therapeutic benefit. Thus, a cocktail therapy of AAVs with immunosuppressors, FOP inhibitors, and/or non-immunogenic ACVR1^R206H^-specific RNA interferences is considered to minimize FOP-associated immune responses.

The clustered regularly interspaced short palindromic repeats (CRISPR)/CRISPR-associated protein 9 (Cas9) system has been developed as a genome-editing tool that can correct DNA mutations. In principle, the CRISPR/Cas9-based adenine base editor (ABE) system is considered to correct the classic *ACVR1^R206H^* mutation (c.617G>A) at the genomic level by directly converting adenine (A) to guanine (G) in the human *ACVR1^R206H^* allele without creating double-stranded DNA breaks [34]. Recent studies demonstrated that a single AAV vector carrying Nme2-ABE and a single guide RNA (sgRNA) enables the correction of disease-causing mutation(s) in vivo [35,36]. Since the systemic delivery of AAV9 serotype transduces HO-causing FAPs in skeletal muscle [9], the AAV9-mediated correction of *ACVR1^R206H^* mutation using the Nme2-ABE/sgRNA might be a promising therapeutic strategy for a permanent cure for HO in FOP patients. However, it is challenging to use an AAV-based CRISPR/ABE system as an in vivo gene therapy for FOP due to several limitations, such as immune reactions against the bacterial nuclease Nme2-ABE, off-target cleavage and mutagenesis, and the induction of chromosomal aberrations. In particular, these concerns become more problematic with long-term AAV-mediated expression. Therefore, further vector improvement to overcome these issues should be considered.

## Figures and Tables

**Figure 1 biomolecules-13-01364-f001:**
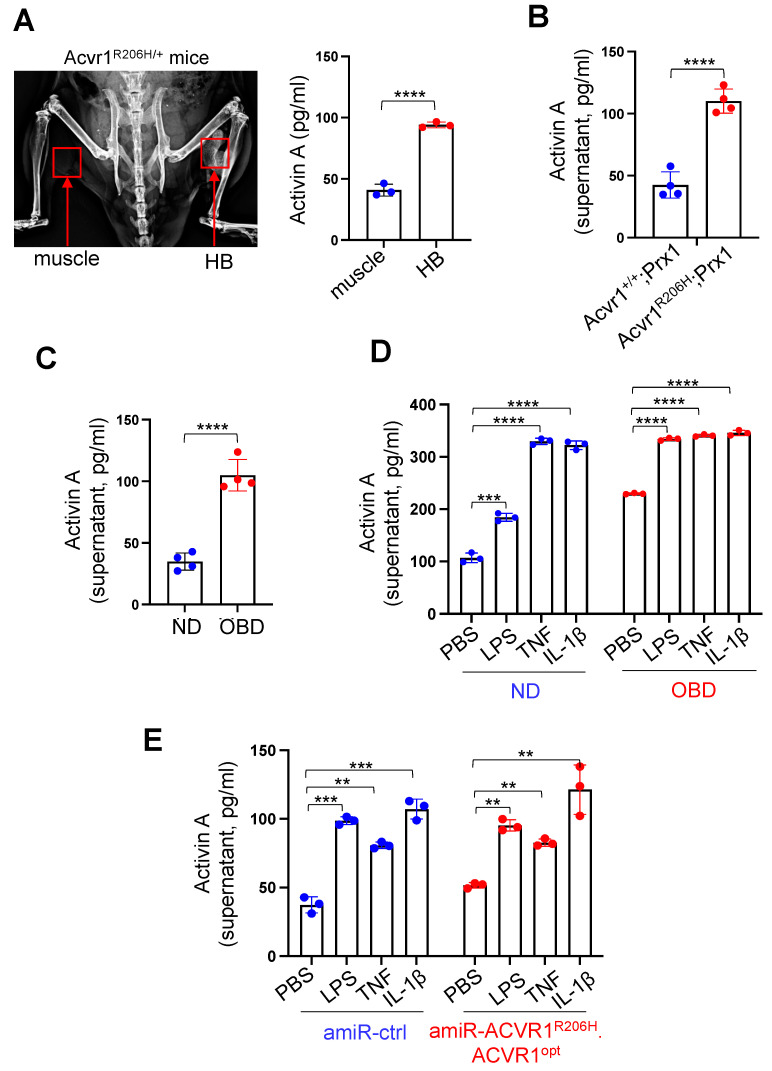
Elevated levels of Activin A in HO tissues of FOP mice. (**A**) Pinch injury was introduced into the right-side of the gastrocnemius muscle of the 8-week-old *Acvr1^R206H/+^* mice, and four weeks later, trauma-induced HO was assessed via radiography (**left**). Alternatively, the protein levels of Activin A in tissue lysates were measured using ELISA (**right**, n = 3). HB: heterotopic bone. (**B**) Protein levels of Activin A in the supernatant of mouse *Acvr1^+/+^;Prx1* (WT) and *Acvr1^R206H^;Prx1* (FOP) BMSCs (n = 4). (**C**,**D**) Mouse (**C**, n = 4) or human (**D**, n = 3) BMSCs were cultured under non-differentiation (ND) or osteoblast differentiation (OBD) conditions in the presence of PBS, LPS, TNF, or IL-1β, and Activin A expression in the supernatant was measured via ELISA. (**E**) Mouse *Acvr1^R206H^;Prx1* BMSCs were transduced using AAV9 carrying control (amiR-ctrl) or *amiR-ACVR1^R206H^*.*ACVR1^opt^* and then cultured under osteogenic conditions in the presence of PBS, LPS, TNF, or IL-1β. Twenty-four hours later, Activin A expression in the supernatant was measured by ELISA (n = 3). **, *p* < 0.01; ***, *p* < 0.001, and ****, *p* < 0.0001 by an unpaired, two-tailed Student’s *t*-test (**A**–**C**) and one-way ANOVA test (**D**,**E**).

**Figure 2 biomolecules-13-01364-f002:**
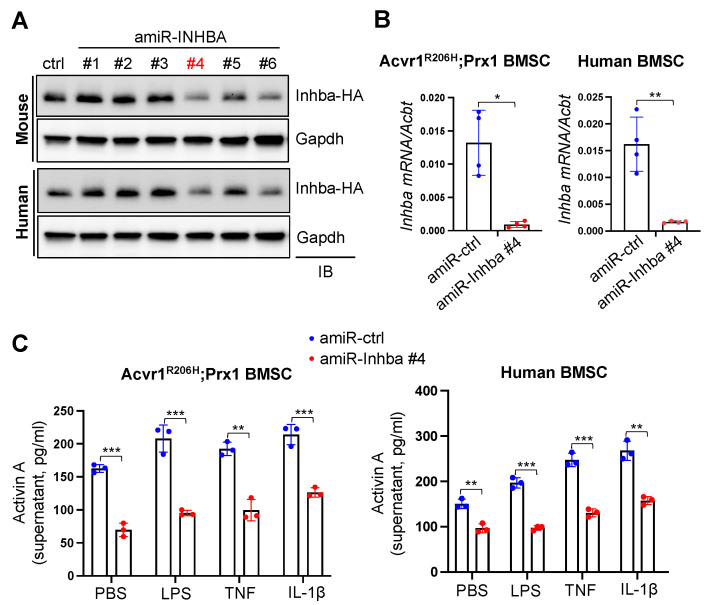
Generation of an Activin A-silencing AAV vector. (**A**) HA-tagged mouse or human Inhba plasmid was transfected into HEK293 cells along with the control (amiR-ctrl) or amiR-Inhba #1–#6 plasmid and then total cell lysates were immunoblotted with the HA antibody. Gapdh was used as a loading control. amiR-Inhba #1–#6 indicate six amiRs targeting the shared coding sequences of mouse and human *Inhba*. Original images can be found in Appendix A. (**B**) Mouse *Acvr1^R206H^;Prx1* BMSCs or human BMSCs were transduced via AAV9 carrying amiR-ctrl or amiR-Inhba #4, and mRNA levels of *Inhba* were assessed using RT-PCR (n = 4). (**C**) AAV-transduced mouse *Acvr1^R206H^;Prx1* BMSCs or human BMSCs were cultured in the presence of PBS, LPS, TNF, or IL-1β, and 24 h later, Activin A expression in the supernatant was measured using ELISA (n = 3). *, *p* < 0.05; **, *p* < 0.01; and ***, *p* < 0.001 via an unpaired two-tailed Student’s *t*-test (**B**) and one-way ANOVA test (**C**).

**Figure 3 biomolecules-13-01364-f003:**
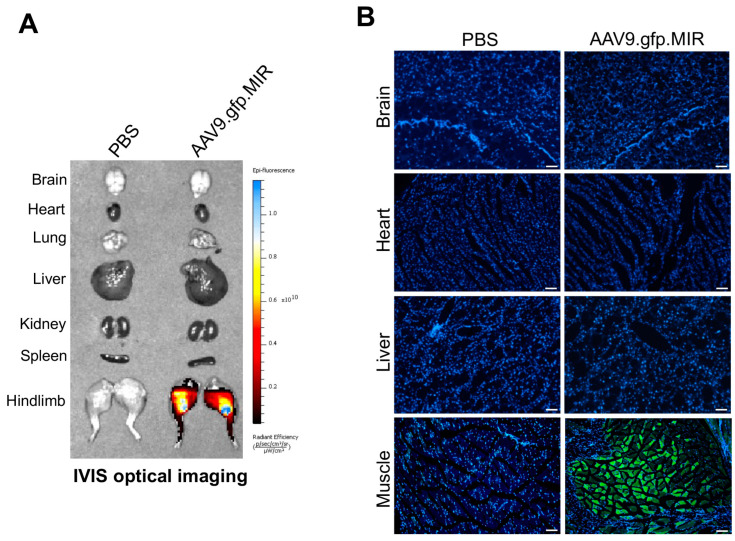
Biodistribution of AAV9.gfp.MIR in mice. A total of 5 × 10^12^ vg/kg of AAV9.gfp.MIR was transdermally (t.d.) injected into 8-week-old *Acvr1^R206H/+^* mice (n = 3), and 2 weeks later, individual tissue distribution of AAVs was assessed via EGFP expression using the IVIS optical imaging system (**A**) or histology on a frozen section of AAV-treated tissues (**B**). Scale bars: 100 μm.

**Figure 4 biomolecules-13-01364-f004:**
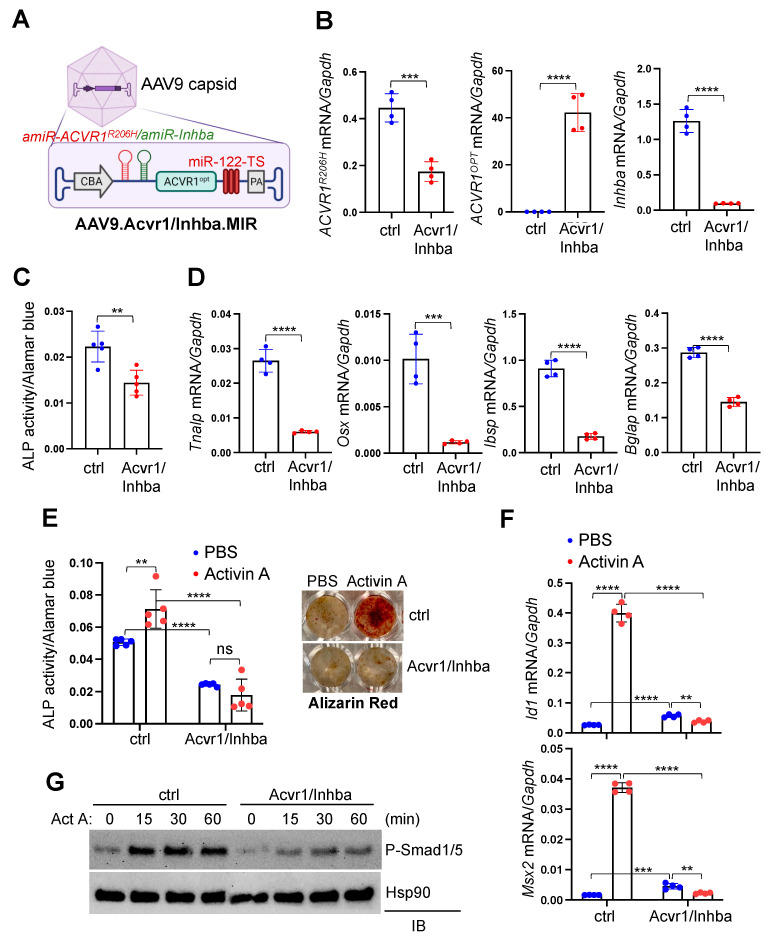
AAV gene therapy suppresses Activin A signaling and osteogenic differentiation of FOP skeletal progenitors. (**A**) A schematic diagram showing AAV vector genome and capsid. AAV vector genome containing the CBA promoter, amiR-ACVR1^R026H^, amiR-Inhba #4 (red), codon-optimized human ACVR1 (ACVR1^opt^), and liver-abundant miR-122 target sequences (TS) was packaged into AAV9 capsid. CBA: chicken β actin; PA: poly A sequences. (**B**–**D**) Mouse *Acvr1^R206H^;Prx1* BMSCs were transduced by AAV9 carrying gfp.MIR (ctrl) or Acvr1/Inhba.MIR and cultured under osteogenic conditions. Six days later, gene expression was measured via RT-PCR and normalized to Gapdh (**B**,**D**, n = 4). Osteogenic differentiation was assessed by ALP activity (**D**, n = 5)**.** (**E**–**G**) AAV-transduced BMSCs were cultured under osteogenic conditions in the presence of PBS or Activin A (50 ng/mL). ALP activity (3 day culture) or alizarin red staining (10 day culture) were performed for early or late osteogenic differentiation, respectively (**D**, n = 5). Activin A-induced expression of *Id1* and *Msx2* was assessed via RT-PCR 24 h post-stimulation (**E**, n = 4). Cells were stimulated with Activin A at different time points and immunoblotted for phospho-Smad1/5. Hsp90 was used as a loading control (**F**). Original images of (**G**) can be found in Appendix A. **, *p* < 0.01; ***, *p* < 0.001, and ****, *p* < 0.0001 with an unpaired two-tailed Student’s *t*-test (**A**–**C**) and one-way ANOVA test (**D**,**E**).

**Figure 5 biomolecules-13-01364-f005:**
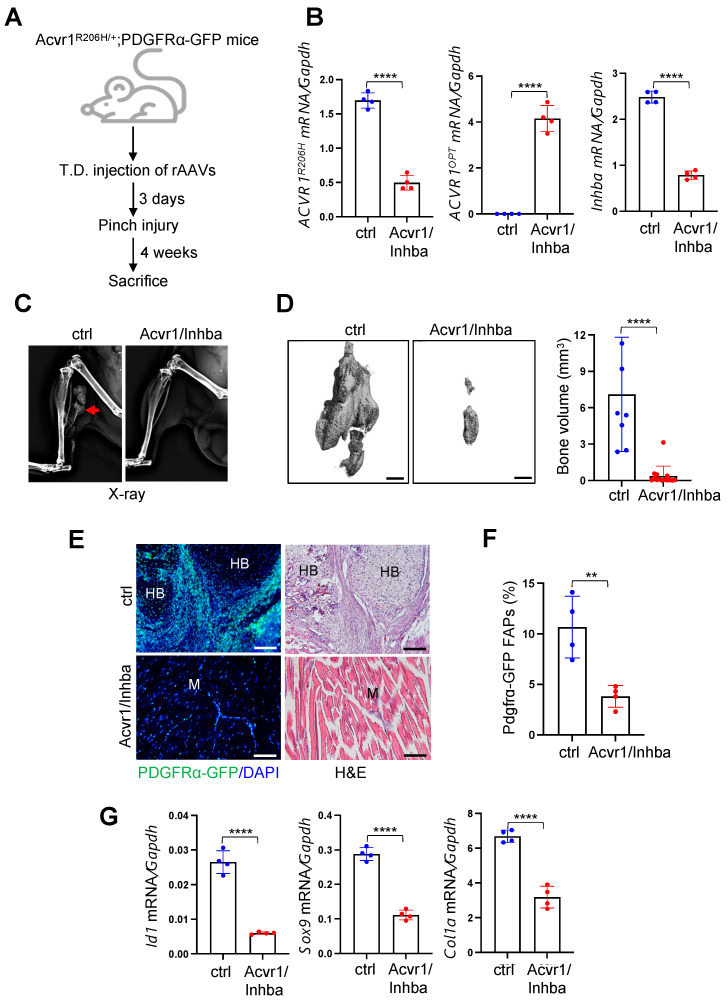
Local delivery of AAV gene therapy prevents trauma-induced HO in FOP mice. (**A**) Diagram of study and treatment methods. A total of 5 × 10^12^ vg/kg of AAV9 carrying gfp.MIR (ctrl) or Acvr1/Inhba.MIR was injected t.d. into the gastrocnemius muscle of 8-week-old *Acvr1^R206H/+^;Pdgfrα-GFP* mice 3 days prior to pinch injury. Four weeks later, *ACVR1^R206H^*, *ACVR1^OPT^*, and *Inhba* expression was assessed using RT-PCR (**B,** n = 4), and HO in the injured muscle was assessed via X-ray (**C**), microCT (**D**), and histology on a frozen section of AAV-treated hindlimbs (**E**). Three-dimensional reconstruction images (**D**, **left**) and quantification of HO volume (**D**, **right**, n = 7~10) are displayed. Red arrow indicates heterotopic bone (**C**). Proliferation of Pdgfrα-GFP-expressing FAPs within HO tissues was visualized using fluorescence microscopy. Scale bars: 100 μm. HB: heterotopic bone; M: muscle. The frequency of GFP^+^ScaI^+^CD31^−^CD45^−^ FAPs in the injured muscle was assessed using flow cytometry (**F**), and Pdgfrα-GFP FAPs were FACS-sorted from the injured muscle and subjected to RT-PCR analysis (**G**, n = 4). **, *p* < 0.01 and ****, *p* < 0.0001 using an unpaired, two-tailed Student’s *t*-test (**B**,**D**,**F**,**G**).

**Figure 6 biomolecules-13-01364-f006:**
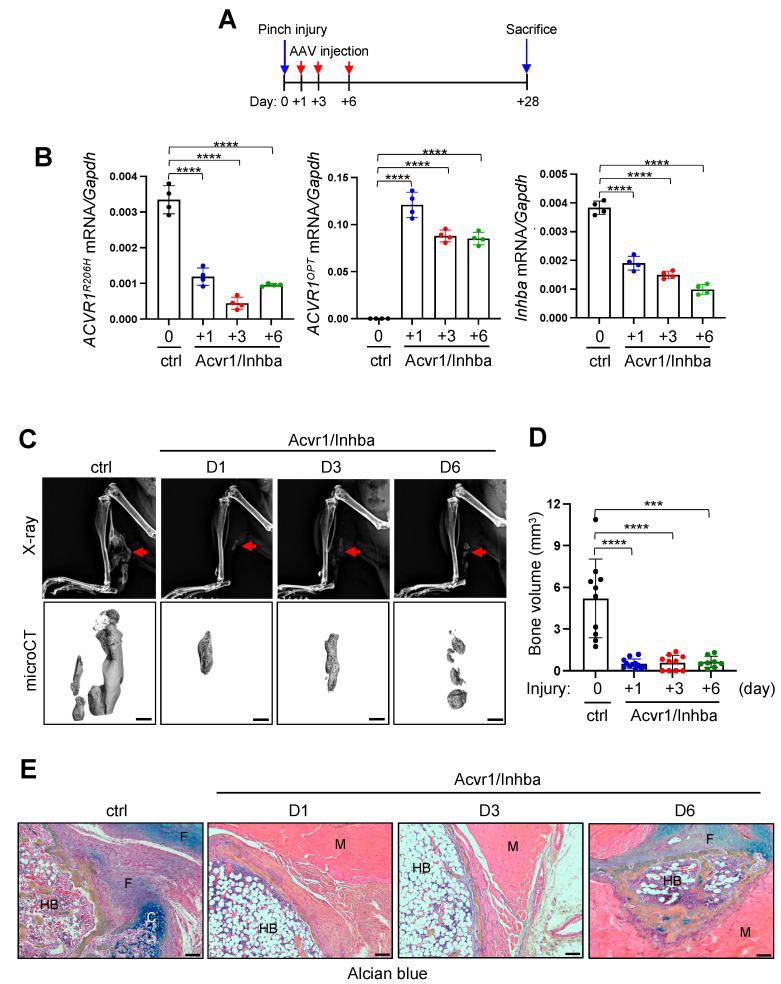
Local delivery of AAV gene therapy suppresses progression of traumatic HO in FOP mice. (**A**) Diagram of study and treatment methods. After pinch injury on the gastrocnemius muscle of 8-week-old *Acvr1^R206H/+^* mice at day 0, t.d. injections of AAV9.*Acvr1/Inhba.MIR* (5 × 10^12^ vg/kg) into the injured muscle were performed on days 1, 3, or 6 post-injury. AAV9.*gfp.MIR* (ctrl) was t.d. injected on the same day of pinch injury. *ACVR1^R206H^*, *ACVR1^OPT^*, and *Inhba* expression were assessed via RT-PCR at 4 weeks post-injury (**B,** n = 4). HO in the injured muscle was assessed using X-ray (**C**, **top**), microCT (**C**, **bottom**), and histology on a paraffin section of AAV-treated hindlimbs (**D**). Three-dimensional reconstruction images (**C**) and quantification of HO volume (**D,** n = 9~10) are displayed. Red arrows indicate heterotopic bones (**C**). Longitudinal sections of the injured muscle was stained with Alcian blue/hematoxylin/orange G (**E**). HB: heterotopic bone; F: fibrotic tissue; C: chondrogenic analgen; M: skeletal muscle. Scale bars: 100 μm. ***, *p* < 0.001 ****, *p* < 0.0001 using a one-way ANOVA test (**A**,**B**,**D**).

## Data Availability

Data supporting the findings of this manuscript are available from the corresponding authors upon request. The raw data are protected and are not available due to data privacy laws. However, the processes described in this study are described in the Appendix A.

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
