# Peer review of "AAV-Mediated Targeting of the Activin A-ACVR1R206H Signaling in Fibrodysplasia Ossificans Progressiva"

_biomolecules, 2023, doi:10.3390/biom13091364_

Round 1

Reviewer 1 Report

This article follows a previous one (ref 13) showing the construction and the effects of a multifunctional AAV that is both silencing the FOP pathogenic molecules Activin A and mutated ACVR1 and expressing the wild type ACVR1. The rationale is that the aberrant activation of SMAD 1-5 signaling is bolcked by restoring wild type receptor expression and its downstream signaling in FOP patients cells.

This strategy showed beneficial effects in in vitro cell based experiments and in vivo in mice carrying the mutated receptor.

In the first part of this article the authors want to stress that chondrogenic/osteogenic cells, in addition to inflammatory immune cells, produce Activin A in the sites where ossification is taking place thus stating that in these sites Activin A is produced by cells that initiate HO and further sustatined by cells that are activated to make HO proceed further. Given this, the authors describe results of the AAV combined effects of inhibition of aberrant signaling and osteogenesis in these types of cells both in in vitro and in in vivo experiments.

As a further proof of the promising application of this therapeutic strategy, the authors describe how the local AAV administration after muscle injury was able to reduce the mass of the heterotopic bone produced after the injury.

The authors have modified, with apparent good results,  the viral construct to overcome liver expression which takes place not only after systemic administration but also after transdermal administration, and therefore they improve safety of the multifunctional AAV.

My main criticismi is about the discussion. A relatively short part of it is devoted to discussing the results and the perspectives of application to clinics of the present version of the multifunctional AAV or its possible further modifications.

I suggest that, at least in the discussion, the issue of the admistration mode is more extensively addressed, being one of the most critical issues in the perspective of a clinical application of this therapeutic strategy. In the article, it is well indicated that, in this set of experiments, the AAV was administerd locally in mice. However, since the positive results of the effects of liver expression silencing were obtained after local administration of the AAV, I think that the systemic effects after local administration should be discussed, also considering what was reported in the previous article published by these authors.

Note. The 2 line sentence that starts at line 400 seems to miss something.

Overall, this article does not have an intrinsic original appearance considering the results previously published by the same authors, but reports some useful advancements of previously described results, that in my opinion are worth to be published.

Author Response

This article follows a previous one (ref 13) showing the construction and the effects of a multifunctional AAV that is both silencing the FOP pathogenic molecules Activin A and mutated ACVR1 and expressing the wild type ACVR1. The rationale is that the aberrant activation of SMAD 1-5 signaling is blocked by restoring wild type receptor expression and its downstream signaling in FOP patients cells. This strategy showed beneficial effects in in vitro cell based experiments and in vivo in mice carrying the mutated receptor.

In the first part of this article the authors want to stress that chondrogenic/osteogenic cells, in addition to inflammatory immune cells, produce Activin A in the sites where ossification is taking place thus stating that in these sites Activin A is produced by cells that initiate HO and further sustained by cells that are activated to make HO proceed further. Given this, the authors describe results of the AAV combined effects of inhibition of aberrant signaling and osteogenesis in these types of cells both in in vitro and in in vivo experiments.

 As a further proof of the promising application of this therapeutic strategy, the authors describe how the local AAV administration after muscle injury was able to reduce the mass of the heterotopic bone produced after the injury. The authors have modified, with apparent good results,  the viral construct to overcome liver expression which takes place not only after systemic administration but also after transdermal administration, and therefore they improve safety of the multifunctional AAV.

Overall, this article does not have an intrinsic original appearance considering the results previously published by the same authors, but reports some useful advancements of previously described results, that in my opinion are worth to be published. My main criticism is about the discussion. A relatively short part of it is devoted to discussing the results and the perspectives of application to clinics of the present version of the multifunctional AAV or its possible further modifications.

-    We thank the reviewer for highlighting the summary and the significance of our manuscript focusing on gene therapy approaches for FOP with advancements in therapeutic efficacy and safety. We believe that addressing these comments in the discussion has improved the revised manuscript. 

I suggest that, at least in the discussion, the issue of the administration mode is more extensively addressed, being one of the most critical issues in the perspective of a clinical application of this therapeutic strategy. In the article, it is well indicated that, in this set of experiments, the AAV was administered locally in mice. However, since the positive results of the effects of liver expression silencing were obtained after local administration of the AAV, I think that the systemic effects after local administration should be discussed, also considering what was reported in the previous article published by these authors.

-    As suggested by the reviewer, the issue of transdermal injection of AAV gene therapy was extensively addressed in the discussion section by adding the paragraph, Since immunization of FOP patients via intramuscular (i.m.) injection often induces muscle trauma and a flare-up, followed by formation of HO lesions, we delivered AAV gene therapy to skeletal muscle via transdermal (t.d.) injection using a hollow microneedle to avoid muscle trauma by i.m. injection. Consistent with our previous study demonstrating that t.d. injection of AAV9 transduces Tie2+ or PDGFRa+ FAP-lineage cells in HO lesions, local delivery of our AAV gene therapy to skeletal muscle via t.d. injection was effective for transduction of FAP-lineage cells and suppressed heterotopic bone formation in the injured muscle of Acvr1R206H/+. Thus, t.d. injection of AAV gene therapy might be a promising approach to treat trauma- and/or flare-up-induced HO in FOP patients. However, since expression of t.d. injected AAV9 was also detected in the liver due to high trans-vascular efficiency of AAV9 capsid, we repressed its expression in the liver via liver-abundant miR-122-mediated degradation since liver function in FOP patients is relatively normal.”

Note. The 2 line sentence that starts at line 400 seems to miss something.

-    We thank the reviewer for correcting this. The sentence was revised to “local delivery of AAV gene therapy to the skeletal muscle before or after pinch injury markedly decreased endochondral bone formation in Acvr1R206H/+mice, suggesting AAV gene therapy as a potent suppressor that can prevent or treat trauma-induced HO in FOP.”

Reviewer 2 Report

The manuscript submitted by Yang et al. focuses on the use of AAV vectors to deliver small interfering sequences to simultaneously inhibit Actvr1 and Inhba. The concept of the viral vector is being used to either prevent or treat heterotopic ossification in the disease FOP. They have used cell culture and in vivo mouse models to test the efficacy of their viral vector. In addition, the vector has additional built-in features to enable normal expression of the wild type codon optimized Acvr1, as well as liver-specific miRNA to avoid side effects in that tissue. The manuscript is very well written and provide very promising data to alleviate the HO in a mouse model. Bone marrow stem cells derived from the mice were also used to show that neutralization of Inhba reduces the osteogenic and chondrogenic potential of the cells. Although this is an excellent manuscript, however, it would benefit from revision according to the following specific comments:

In the abstract and introduction, the statement that there is no drug available for FOP should be re-considered. The FDA has recently approved Sohonos (palovarotene) for HO and FOP.

Line 51, the rAAVs is first mentioned and should be defined. Line 53, use the abbreviation only.

Line 128-129, The description of the MBL should be described, or a reference to it provided.

Line 131, typo xie?

For Figure 1 D and E, the statistical analysis should be indicated on the panels. The corresponding text should also specify if there are differences between the control and treated cells. Was there a difference expected between ctrl and the amir-acvrr206.acvr1opt? Because they look almost identical.

For panel B, western blot for HA-tagged Inhba, it should be clarified that the samples were total cell lysates (not media).

For Figure 2, it is unclear what the amiR-INHBA #1 to #6 relate to. This should be clearly stated in the figure legend. It should also be specified that the selected amiR-INHBA has full homology to mouse and human IHNBA.

Also, when comparing INHBA levels presented in figure 2 D and Figure 1E, they seem to differ and cells being less responsive to the inflammatory agents. Any explanation for the biological variability?

Line 283, What was the rationale for using PMA to stimulate Ihnba expression?

Line 284, The expression of ACVR1 driven by the AAV is not demonstrated yet in the figures presented thus far. So this should probably be left out from this paragraph as it appears only in figure 4.

Line 288, The rationale presented for using the miR-122 sequence in the AVV9, would be to repress potential expression of the Ihnba and Acvr1 miR in the liver, with the idea of reducing potential non-tissue specific effects. However, the data presented in section 3.3 and figure 3, was for transdermal 'local' injections into the muscle, mainly. Would it be expected to get liver expression using this route of administration, or not? This would be likely the case if the AVV9 was to be injected i.v. Further, to fully demonstrate the efficacy of the miR122, a side-by-side comparison of the injections of AVV9 carrying, or not, the miR122 would have been required. Without providing this data, one cannot conclude of the necessity for miR122 in the construct design. The discussion should also be modified accordingly when describing the ‘detargeting’ of the liver.

Line 312-314, For testing the osteogenic differentiation in the AcvrR206H BMSCs, the results of figure 4B and C point to a decreased expression of 4 bone gene markers. Were the expression levels of the 4 genes higher in these BMSCs relative to WT BMSCs? So the effect of the AAV9 would actually 'normalize' expression and activity?

The data presented in figure 4, panels D and E, should show the statistical comparisons between ctrl and acvr1/inhba effects. This is important to show the baseline and repressed levels of gene expression. It looks like there might be a slight increase id Id1 and Msx2 in the infected cells (blue symbols).

As explained in the text, the legend to Figure 6 should clearly specify that the injections were done at d+1, D=3, OR D+6, not receiving 3 consecutive injection. Also, the labeling for panels C and E should be changed and rather indicated ctl (non-injected) instead of D0. The current labeling indicating 0 could be mis-interpreted as the baseline phenotype before treatment.

Line 402-403, This sentence is confusing, especially 'expression in the liver with no FOP pathology'? What does this mean?

Author Response

The manuscript submitted by Yang et al. focuses on the use of AAV vectors to deliver small interfering sequences to simultaneously inhibit Actvr1 and Inhba. The concept of the viral vector is being used to either prevent or treat heterotopic ossification in the disease FOP. They have used cell culture and in vivo mouse models to test the efficacy of their viral vector. In addition, the vector has additional built-in features to enable normal expression of the wild type codon optimized Acvr1, as well as liver-specific miRNA to avoid side effects in that tissue. The manuscript is very well written and provide very promising data to alleviate the HO in a mouse model. Bone marrow stem cells derived from the mice were also used to show that neutralization of Inhba reduces the osteogenic and chondrogenic potential of the cells. Although this is an excellent manuscript, however, it would benefit from revision according to the following specific comments:

-    We thank the reviewer for highlighting the significance of our manuscript. We believe that addressing these comments has improved the revised manuscript. 

In the abstract and introduction, the statement that there is no drug available for FOP should be re-considered. The FDA has recently approved Sohonos (palovarotene) for HO and FOP.

-    As suggested by the reviewer, the sentence, “Currently, there are no effective preventions or therapies for FOP.”, was removed from the revised manuscript.

Line 51, the rAAVs is first mentioned and should be defined. Line 53, use the abbreviation only.

-   We thank the reviewer for correcting this.

Line 128-129, The description of the MBL should be described, or a reference to it provided.

-    As suggested by the reviewer, a short description and reference were added to the revised manuscript.

Line 131, typo xie?

-    We thank the reviewer for correcting this.

For Figure 1 D and E, the statistical analysis should be indicated on the panels. The corresponding text should also specify if there are differences between the control and treated cells. Was there a difference expected between ctrl and the amir-acvrr206.acvr1opt? Because they look almost identical.

-     As suggested by the reviewer, the statistical analysis and corresponding text specifying the differences were added to the revised manuscript.

-     The reviewer is correct. The responsiveness of FOP BMSCs to pro-inflammatory stimuli that upregulate Activin A expression was not altered by the amir-acvrr206.acvr1opt.

For panel B, western blot for HA-tagged Inhba, it should be clarified that the samples were total cell lysates (not media).

-    We thank the reviewer for pointing this out. The samples were total cell lysates, not the supernatant, which was clarified in the revised manuscript.

For Figure 2, it is unclear what the amiR-INHBA #1 to #6 relate to. This should be clearly stated in the figure legend. It should also be specified that the selected amiR-INHBA has full homology to mouse and human IHNBA.

-    As suggested by the reviewer, the sentence, “six amiRs targeting shared coding sequences of mouse and human Inhba (amiR-Inhba #1-#6) were designed to silence expression of both mouse and human Activin A” was added to the revised manuscript.

-    The sentence, “These results demonstrated that amiR-Inhba #4 and #6 with full homology to mouse and human IHNBA effectively silenced expression of both mouse and human Inhba.” was added to the revised text.

Also, when comparing INHBA levels presented in figure 2 D and Figure 1E, they seem to differ and cells being less responsive to the inflammatory agents. Any explanation for the biological variability?

-   We agree with the reviewer’s concern about biological variability in the responsiveness of primary Acvr1R206H;Prx1 BMSCs to pro-inflammatory stimuli. To obtain primary BMSCs, we crushed the femurs and tibias of 4-week-old Acvr1(R206H)Fl;PRRX1-Cre mice and then, total cells (bone marrow cells, stromal cells, and osteoblasts) were harvested from the crushed bones. After the removal of red blood cells using RBC lysis buffer, they were plated onto a cell culture dish. Two weeks after the culture in the growth medium, total adherent cells were harvested and frozen for storage. This BSMC isolation protocol shows experimental limitations, such as heterogenecity of primary BMSCs and contamination with other cell populations when harvested from heterotopic bones in Acvr1(R206H)Fl;PRRX1-Cre mice, which may cause this variability.

Line 283, What was the rationale for using PMA to stimulate Ihnba expression?

-    We thank the reviewer for pointing this out. A previous study has reported that PMA treatment of human BMSCs upregulated the expression of Activin A. However, since the mitogen PMA is not related to inflammation, we removed PMA stimulation data from the revised Figure 2D.

Line 284, The expression of ACVR1 driven by the AAV is not demonstrated yet in the figures presented thus far. So this should probably be left out from this paragraph as it appears only in figure 4.

-    As suggested by the reviewer, along with the diagram in Figure 2A, this sentence was moved to the Figure 4A and the corresponding text in the revised manuscript.

Line 288, The rationale presented for using the miR-122 sequence in the AVV9, would be to repress potential expression of the Ihnba and Acvr1 miR in the liver, with the idea of reducing potential non-tissue specific effects. However, the data presented in section 3.3 and figure 3, was for transdermal 'local' injections into the muscle, mainly. Would it be expected to get liver expression using this route of administration, or not? This would be likely the case if the AVV9 was to be injected i.v. Further, to fully demonstrate the efficacy of the miR122, a side-by-side comparison of the injections of AVV9 carrying, or not, the miR122 would have been required. Without providing this data, one cannot conclude of the necessity for miR122 in the construct design. The discussion should also be modified accordingly when describing the ‘detargeting’ of the liver.

-    We thank the reviewer for pointing this out. Although we could not provide a side-by-side comparison of AAV9.egfp and AAV9.egfp.MIR in mice, our previous study using the IVIS optical imaging system demonstrated a robust expression of t.d. injected AAV9.egfp in the liver due to high trans-vascular efficiency of the AAV9 capsid (Yang et al., Nat. Commun., 2022). We, therefore, repressed AAV9’s expression in the liver via liver-abundant miR-122-mediated degradation to improve the safety of the AAV gene therapy. For the rationale using the liver-detargeting AAV9, this sentence and the reference were added to the revised manuscript.

-    We agree with the reviewer’s suggestion that our liver-detargeting AAV9 would be also useful for systemic delivery since i.v. injection of AAV9 transduces the liver. This was addressed in the discussion section of the revised manuscript.

Line 312-314, For testing the osteogenic differentiation in the AcvrR206H BMSCs, the results of figure 4B and C point to a decreased expression of 4 bone gene markers. Were the expression levels of the 4 genes higher in these BMSCs relative to WT BMSCs? So the effect of the AAV9 would actually 'normalize' expression and activity?

-    We thank the reviewer for pointing this out. To test whether AAV treatment could normalize osteogenic gene expression and osteoblast activity of FOP BMSCs to those of WT BMSCs, primary BMSCs were isolated from Acvr1R206H;Prx1 mice or Acvr1+/+;Prx1 mice and treated with ctrl or AAV9.Acvr1/Inhba. We found that cell proliferation rate of Acvr1R206H;Prx1 BMSCs was significantly higher than that of Acvr1+/+;Prx1 BMSCs, resulting in a big difference in cell numbers during AAV treatment and osteoblast differentiation. This disparity in cell numbers highly affected osteogenic potentials of FOP and WT BMSCs, which made our comparison study challenging.

The data presented in figure 4, panels D and E, should show the statistical comparisons between ctrl and acvr1/inhba effects. This is important to show the baseline and repressed levels of gene expression. It looks like there might be a slight increase of Id1 and Msx2 in the infected cells (blue symbols).

-    As suggested by the reviewer, the statistical comparisons between ctrl and acvr1/inhba effects were added to the revised figure 4.

-    The reviewer is correct. there was a slight increase in Id1 and Msx2 expression in PBS-treated cells, which is statistically significant.

As explained in the text, the legend to Figure 6 should clearly specify that the injections were done at d+1, D=3, OR D+6, not receiving 3 consecutive injection. Also, the labeling for panels C and E should be changed and rather indicated ctl (non-injected) instead of D0. The current labeling indicating 0 could be mis-interpreted as the baseline phenotype before treatment.

-    As suggested by the reviewer, Figure 6 legend and the labeling for Figure 6C and E were revised.

Line 402-403, This sentence is confusing, especially 'expression in the liver with no FOP pathology'? What does this mean?

-    As suggested by the reviewer, this sentence was revised to “insertion of target sequences for liver abundant miR-122 into AAV vector genome enabled repressing AAV9’s expression in the liver.”